# Primary care pharyngotonsillitis complications following absent or deferred antibiotic treatment across the COVID 19 pandemic

Ailiana Santosa [1] ✉, Julius Collin[2], Elin Dahlén[3,4], Anders Lignell[5], Maria Furberg [3,6], Anders Ternhag [2,7], Rickard Ljung [3,8] & Fredrik Nyberg [1]

## Abstract

**Background** Antibiotic prescribing for pharyngotonsillitis in primary healthcare significantly influences patient outcomes, particularly concerning the risk of complications. This study examines trends in antibiotic use and the associated risk of complications before, during and after the pandemic.
**Methods** This study used large-scale, register-based data linking primary healthcare records with national and regional registers. The study included individuals aged 12 and older from the Stockholm and Västra Götaland regions, representing about 40% of the Swedish population. We identified 295,972 cases of pharyngotonsillitis between 1 January 2018 and 30 December 30, 2023, applying a 180-day washout period to ensure distinct episodes. Logistic regression models estimated adjusted odds ratios (aORs) with 95% confidence intervals (CIs) for complications within 30 days, comparing patients who received antibiotics to those who did not.
**Results** Antibiotic prescriptions temporarily decrease during the pandemic, followed by a partial rebound, with penicillin remaining the preferred antibiotic. Complications, with peritonsillar abscess being the most common, were more frequent in patients receiving antibiotics (1.75%) compared to untreated individuals (0.43%). Among treated patients, those prescribed penicillin had fewer complications (1.62%) than those given other antibiotics (2.87%). After adjusting for sociodemographics, comorbidities, primary care visits, and vaccination status, the risk for complications is lower for untreated patients (aOR 0.24, 95% CI 0.22–0.26).
**Conclusions** The COVID-19 pandemic impacted antibiotic prescribing patterns in Swedish primary care, with a substantial reduction in overall antibiotic prescribing during the pandemic. Penicillin's continued use as a first-line therapy appears well-justified, given its lower associated risk of complications.

## Plain language summary

Throat infections are common in primary care, and antibiotics are often prescribed when they are not needed. How antibiotics are used may affect the risk of health problems. We examine how antibiotic prescribing for throat infections in Sweden changed before, during and after the COVID-19 pandemic, and whether this is linked to complications. Using health care records from 300,000 visits in two Swedish regions, we studied antibiotic use and complications. We find that antibiotic use decreased during the pandemic and increased again. Patients who receive antibiotics had more complications than those who did not, and penicillin was associated with fewer complications than other antibiotics. This highlights the importance of careful antibiotic prescribing and supports penicillin as a first choice for treatment.

Pharyngotonsillitis, an inflammation of the pharynx and tonsils, is commonly caused by viral or bacterial infections, with *Streptococcus pyogenes* (group A streptococci, GAS) being the most significant bacterial pathogen. Treatment guidelines[1–5] generally recommend antibiotics only for GAS-related infections. Since 2001, the Swedish Medical Products Agency and Swedish Strategic Programme for the Rational Use of Antimicrobial Agents

and Surveillance of Resistance (STRAMA)[3,6] have recommended using the Centor scoring system to identify patients at higher risk of GAS infection. For patients with a Centor score of 3−4 and a positive GAS rapid antigen test, penicillin V is the recommended treatment[1].

During the COVID-19 pandemic, viral infections such as influenza and respiratory syncytial virus, as well as bacterial respiratory infections like

otitis media, declined notably across Europe[7–9] and the US[10]. In the UK, a significant reduction in pharyngotonsillitis episodes was observed among children with recurrent pharyngotonsillitis during lockdown[11]. Similarly, a retrospective study in the US (2019–2021) reported a sharp decline in tonsil-related diagnoses in April 2020, though these later rebounded to exceed those seen pre-pandemic[12]. Reduced healthcare visits during the pandemic also resulted in less antibiotics being prescribed and dispensed across Europe[13]. A systematic review of 81 studies[14] found a 37% decline in healthcare visits early in the pandemic, with a greater reduction among individuals with less severe illness. In Sweden, primary care consultations decreased by around 12%, particularly among patients over 65, with no observed gender differences[15]. In the US, outpatient antibiotic prescriptions dropped by 40% between January and May 2020 compared to 2017–2019[16]. Conversely, in the UK, primary care antibiotic prescriptions increased during the pandemic despite fewer appointments from April to August 2020 compared to 2019[17]. In Sweden, the decline in antibiotic prescription in 2020 marked the largest reduction in 20 years[18].

The impact of antibiotic treatment on complications from upper respiratory tract infections (URTIs) and urinary tract infections (UTIs) has been well-documented[19,20]. Antibiotic prescriptions reduced return visits for patients with a positive Rapid Antigen Detection Test (RADT) for GAS, while those with a negative RADT tend to have more return visits[21]. In Sweden, the incidence of uncomplicated bacterial infections, including otitis media, acute rhinosinusitis, and acute pharyngotonsillitis dropped by 40% in 2020 compared to 2019, particularly among patients aged 0–19, alongside declines in antibiotic prescriptions and complications from URTIs, such as acute mastoiditis and peritonsillitis[22]. Pre-pandemic studies from Sweden[23] and the UK[24] found that URTIs rarely cause complications, and antibiotic mainly shortens the time with symptoms and help prevent the spread of infection. Nevertheless, there have been anecdotal reports of an increasing proportion of complications following pharyngotonsillitis during and after the COVID-19 pandemic, alongside a major upsurge of invasive group A streptococci across Europe[25] and other regions post-pandemic[26,27]. However, it remains unclear whether these trends persisted during the continued pandemic or after the pandemic and how they relate to the risk of complications following pharyngotonsillitis. Using nationwide healthcare data, we examine antibiotic prescribing for pharyngotonsillitis across the pre-pandemic, pandemic and post-pandemic periods. We show that prescribing declines during the pandemic and partially rebounds afterwards, while penicillin V remains the predominant treatment. Complications are more frequent among patients receiving antibiotics than among those not treated, and penicillin V is associated with fewer complications than other antibiotics. These findings suggest that reduced antibiotic use during the pandemic is not associated with increased complications following pharyngotonsillitis.

## Method
This study was conducted in accordance with the Declaration of Helsinki and complies with relevant ethical regulations. Ethical approval for the research was obtained from the Swedish Ethical Review Authority (approval no. 2020-01800, with subsequent amendments).

### Data sources
This cohort study is part of the RECOVAC study (Register-based large-scale national population study to monitor COVID-19 vaccination effectiveness and safety) within the SCIFI-PEARL (Swedish Covid-19 Investigation for Future Insights—a Population Epidemiology Approach using Register Linkage) project, described in detail elsewhere[28], and expanded to cover the entire Swedish population. We linked individual data from multiple Swedish national and regional registers. Diagnoses from outpatient specialist visits and inpatient care came from the National Patient Register, while primary care diagnoses were obtained from regional primary healthcare registers for Stockholm Region and Västra Götaland Region (the two largest regions in Sweden, approximately 40% of the population). Detailed records of healthcare contacts, diagnoses, and prescribed medications were obtained from the National Patient Register (2015 onward) and the National Prescribed Drug Register (2018 onward). Sociodemographic data, including education, marital status, and country-of-birth, were acquired from Statistics Sweden (covering 2015–2019), and COVID-19 vaccination data were retrieved from the national vaccination register.

### Study population and index infection episodes
The study included individuals aged 12 and older residing in Stockholm Region and Västra Götaland Region who had at least one primary care visit between 1 January 2018 and 30 December 2023. To identify unique episodes of pharyngotonsillitis for the same patient and exclude recurrent or chronic cases, a 180-day washout period was applied prior to each primary care visit. Thus, for an episode to be counted as a new episode, at least 180 days prior to that primary care visit had to be free of any records of pharyngotonsillitis in primary, specialist outpatient or inpatient care.

### Study period, exposures and follow-up risk window
The study period spanned from 1 January 2018 to 31 January 2024, covering both the initial diagnosis of pharyngotonsillitis and the occurrence of any complications within one month of the pharyngotonsillitis diagnosis. Antibiotic treatment exposure was defined as a dispensed prescription for an antibiotic used for upper respiratory infection, prescribed within 2 days of each index date of a pharyngotonsillitis episode. Complications were assessed for a follow-up of 30 days after the pharyngotonsillitis diagnosis. The study period was divided into pre-pandemic (1 January 2018—31 January 2020), pandemic (1 February 2020—28 February 2022), and post-pandemic (1 March 2022—30 January 2024), defined according to the date of the index pharyngotonsillitis diagnosis.

### Outcome
Infectious complications were identified based on ICD-10 in primary care, outpatient specialist care and hospital inpatient care. They included peritonsillar abscess (J36), retro-parapharyngeal abscess (J39.0), necrotizing fasciitis (M72.6), invasive Group A streptococcal disease (A40.0, A40.3, A40.9) and rheumatic fever (I00) (Supplementary Table 1).

### Exposures
Antibiotic prescriptions for pharyngotonsillitis were identified using ATC codes (Supplementary Table 2) and coded as "yes" or "no", with the reference group being those who *received* antibiotic treatment. We defined deferred antibiotics if prescriptions were dispensed between days 3 and 7 after the index diagnosis. Antibiotic type was further dichotomised as phenoximethylpenicillin (penicillin-V) or other antibiotics, with penicillin-V serving as the reference group.

### Covariates
The following covariates were used for adjustment and stratification: age (12–24, 25–39, 40–69, 70+), sex, country of birth (Nordic/other countries), marital status (married/not married), education (primary, secondary, tertiary less than three years, tertiary three years or longer), number of primary care visits from the index date to outcome occurrence (one, two, three, four or more), vaccination status (none, one dose, two doses, booster), the Charlson Comorbidity Index (CCI) based on diagnoses in the three years preceding the index date (categorized 0,1,2-3,4-5 and ≥6) (Supplementary Table 3) as well as immunosuppression status (autoimmune disease and immunosuppressive treatment) (Supplementary Table 4).

### Statistical analysis
Descriptive analyses were conducted to examine trends in antibiotic treatment and type of antibiotic use over the pandemic period. Standardized mean differences (SMDs) were calculated for each baseline characteristic to quantify differences between antibiotic-treated and untreated groups; values ≥ 0.1 were considered to indicate meaningful imbalance. Logistic regression was used to estimate odds ratios (ORs) with 95% confidence

intervals (CIs) for complications within 30 days, comparing those who did not received antibiotics to those who did (reference group) among pharyngotonsillitis patients. Given the low incidence of complications ( < 2%), ORs were considered appropriate as they closely approximate risk ratios in this setting[29]. Apart from the crude unadjusted model, an adjusted model included adjustment for sex, age, sociodemographic factors, number of primary care visits, CCI, and immunosuppression status. Similar regressions were done only among antibiotic-treated patients, comparing those prescribed other antibiotics to patients treated with penicillin-V (reference group). Subgroup analyses were conducted by sex and age. Results are presented as crude and adjusted odds ratios (aORs) with 95% confidence intervals (95%CI). We conducted a sensitivity analysis extending the exposure window for antibiotic prescriptions from 2 to 5 days after the index diagnosis to assess the robustness of the exposure definition. Statistical analyses were performed in SAS 9.4 and STATA 18.

## Results

### Characteristics of study participants

The study includes 295,972 pharyngotonsillitis patients identified from the Stockholm and Västra Götaland regions between 1 January 2018 and 31 December 2023 (Table 1). Of these, 37.6% were male, with the predominant age group being 25–39 years (39.2%). Most patients had tertiary education (63.7%), were not married (70.1%), were Nordic-born (78.5%), had received a COVID-19 booster (51.6%), and had only one primary care visit (74.1%), with only 6.1% of patients with ≥3 visits. The Charlson Comorbidity Index showed that 81.2% had no comorbidities. In addition, 8.4% had autoimmune disease, and 9.4% had glucocorticoid treatment. Most baseline characteristics were well balanced (SMD < 0.1), with moderate imbalances only for age (SMD = 0.13–0.17) and comorbidity burden (SMD = 0.11–0.12).

### Trend in antibiotic treatments

Table 2 shows trends in antibiotic treatment and types of antibiotics used among pharyngotonsillitis patients across three periods. The proportion of patients receiving antibiotics decreased significantly during the pandemic (39.9%) compared to pre-pandemic levels (53.4%), then increased post-pandemic (51.3%) to a level similar to pre-pandemic. Monthly prescribing trends (Supplementary Fig. 1) confirm this pattern, showing clear seasonal peaks pre-2020, a sharp decline at the onset of the COVID-19 pandemic, and only a partial rebound from 2022 onwards. Despite these changes in overall prescribing, penicillin-V remained the most prescribed antibiotics, with consistently high usage across all periods. Patients with and without antibiotic treatment showed similar characteristics (Table 1), but with fewer treated patients aged 70 or older (2.1%), compared to untreated patients (4.4%). Additionally, a high CCI score (over 6) was less common among treated patients (5.6%) than untreated patients (7.8%).

Characteristics of pharyngotonsillitis patients who received antibiotic treatment across periods highlight that, post-pandemic, a higher proportion of patients aged 25–39 received antibiotics, while the proportion of younger patients (12–24 years) receiving antibiotics decreased compared to pre-pandemic levels (Supplementary Table 5). Additionally, there was an increase in antibiotic treatment among patients with autoimmune diseases and those on glucocorticoid therapy across periods.

### Complications in pharyngotonsillitis patients

Complications occurred in 1.75% of patients who received antibiotics, compared to 0.43% of those who did not receive antibiotics (Table 3). The overwhelmingly most common complication was peritonsillar abscess, occurring in 1.72% of patients receiving antibiotics and 0.41% of untreated patients. Other types of complications were very rare across all groups. In terms of antibiotic type, 1.62% of patients treated with penicillin-V experienced complications, compared to 2.87% of those who received other antibiotics. Peritonsillar abscess was the most frequent complication in both groups. Rare complications, like retropharyngeal and parapharyngeal abscesses, occurred slightly more often with other antibiotics, while necrotizing fasciitis and bacteremia were exceedingly rare.

Complication among patients receiving antibiotics varied across periods, with the highest proportion observed during the pandemic at 2.58%, predominantly due to peritonsillar abscesses (Supplementary Table 6). Pre-pandemic and post-pandemic rates were lower, at 1.38% and 1.92%, respectively, with peritonsillar abscesses also being the predominant complication. Untreated patients consistently lower proportion of complications across all periods, with an increase observed from 0.38% pre-pandemic to 0.51% post-pandemic, primarily due to peritonsillar abscesses.

### Associations between antibiotic treatments and complications within 30 days

Table 4 presents the unadjusted and adjusted odds ratios of complication within 30 days across the study period. The adjusted odds ratio (aOR) of experiencing any complication within 30 days was lower in untreated patients compared to antibiotic-treated patients, with an aOR of 0.24 (95% CI 0.22–0.27). This trend was consistent across all pandemic periods, with the lowest odds ratios observed during the pandemic (aOR 0.16, 95%CI 0.13–0.19). For specific complications (Table 5), the odds ratio of developing a peritonsillar abscess, the most common complication, was 0.23 (95% CI 0.21–0.26) for untreated patients compared to antibiotic-treated patients.

Among antibiotic-treated patients, those prescribed other antibiotics had higher odds of complications within 30 days compared to patients treated with penicillin-V (aOR 1.84, 95% CI 1.66–2.04) (Table 4). For specific complications, such as retropharyngeal or parapharyngeal abscess, the odds ratio was even higher (3.98, 95% CI 2.32–6.82).

Patients without antibiotic treatment had a significantly reduced risk of complications among pharyngotonsillitis patients that was consistent across all age groups, with the lowest odds ratio observed in the oldest age group 70+ (aOR 0.17, 95% CI 0.10–0.29) (Table 6). Similarly, the reduced risks were observed in both males and females. Among patients treated with antibiotics, those receiving other antibiotic had a higher odds ratio of complications compared to those treated with penicillin-V, and this was also regardless of age or sex.

Results from sensitivity analyses showed extending the exposure window to 5 days slightly increased the number of antibiotic prescriptions (from 52.0% to 53.3%) (Supplementary Table 7), while complication rates remained unchanged, confirming the robustness of the main findings (Supplementary Table 8).

## Discussion

This study provides a comprehensive analysis of antibiotic prescribing trends and the occurrence of complications among pharyngotonsillitis patients living in Stockholm and Västra Götaland regions across different periods (pre-pandemic, pandemic, and post-pandemic). These two populous regions in Sweden represent approximately 40% of the national population. The main findings indicate: (1) a temporary decrease in antibiotic prescriptions during the pandemic, followed by a partial rebound afterward, with penicillin-V consistently being the preferred antibiotics; (2) complications were more frequent in patients receiving antibiotics (1.75%) compared to those not treated (0.43%), and patients treated with penicillin-V had fewer complications (1.62%) than those treated with other antibiotics (2.97%); and (3) after adjusting for sociodemographics, comorbidities, primary care visits, and vaccination status, the odds ratio of experiencing complications within 30 days was lower (aOR 0.24, 95%CI 0.22–0.26) for those who did not receive antibiotic treatment.

The COVID-19 pandemic significantly impacted antibiotic prescribing patterns for common infections[17], including pharyngotonsillitis, in primary care settings. This study observed a marked decline in antibiotic prescriptions during the COVID-19 pandemic, with a rebound afterward, consistent with prior research[30–32]. Despite these fluctuations, penicillin-V remained the first-choice antibiotic for treating pharyngotonsillitis throughout the study period, with only 10% of patients being prescribed other antibiotics. Penicillin-V continues to be the first-choice antibiotic for treating group A beta-hemolytic streptococcal (GABHS) pharyngotonsillitis

**Table 1 | Characteristics, including sociodemographics, Charlson comorbidity index, primary care visits, COVID-19 vaccination and immunosuppression status, among pharyngotonsillitis patients living in Stockholm and Västra Götaland regions in Sweden between 1 January 2018 and 31 December 2023, overall and by antibiotic treatment (yes/no)**

| Characteristics | Overall n = 295,972 | Antibiotic treatment | | |
|---|---|---|---|---|
| | | Yes (n = 154,021) | No (n = 141,951) | SMD |
| Sex, male, n (%) | 111,407 (37.6%) | 57,544 (37.4%) | 53,863 (37.9%) | 0.01205 |
| Age group, n (%) | | | | |
| 12−24 | 87,133 (29.4%) | 45,161 (29.3%) | 41,972 (29.6%) | 0.17308 |
| 25−39 | 116,019 (39.2%) | 64,827 (42.1%) | 51,192 (36.1%) | |
| 40−69 | 83,318 (28.2%) | 40,780 (26.5%) | 42,538 (30.0%) | |
| >=70 | 9502 (3.2%) | 3253 (2.1%) | 6249 (4.4%) | |
| Education level, n (%) | | | | 0.04219 |
| Primary school | 75,904 (25.6%) | 38,911 (25.3%) | 36,993 (26.1%) | |
| Secondary school | 31,788 (10.7%) | 16,014 (10.4%) | 15,774 (11.1%) | |
| Tertiary school<3 years | 79,771 (27.0%) | 42,753 (27.8%) | 37,018 (26.1%) | |
| Tertiary school >=3 years | 108,509 (36.7%) | 56,343 (36.6%) | 52,166 (36.7%) | |
| Marital status, n (%) | | | | 0.07008 |
| Not married | 207,405 (70.1%) | 105,564 (68.5%) | 101,841 (71.7%) | |
| Married/register partner | 88,567 (29.9%) | 48,457 (31.5%) | 40,110 (28.3%) | |
| Country of birth, n (%) | | | | 0.06413 |
| Nordic countries | 232,387 (78.5%) | 118,989 (77.3%) | 113,398 (79.9%) | |
| Other countries | 63,585 (21.5%) | 35,032 (22.7%) | 28,553 (20.1%) | |
| Vaccination status, n (%) | | | | 0.06006 |
| None | 44,178 (14.9%) | 24,047 (15.6%) | 20,131 (14.2%) | |
| 1 dose | 8482 (2.9%) | 4570 (3.0%) | 3912 (2.8%) | |
| 2 doses | 90,635 (30.6%) | 48,067 (31.2%) | 42,568 (30.0%) | |
| Booster (3rd) dose | 152,677 (51.6%) | 77,337 (50.2%) | 75,340 (53.1%) | |
| Number of primary care visits, n (%) | | | | 0.06446 |
| One visit | 219,273 (74.1%) | 112,090 (72.8%) | 107,183 (75.5%) | |
| Two visits | 58,640 (19.8%) | 31,822 (20.7%) | 26,818 (18.9%) | |
| Three visits | 14,247 (4.8%) | 7929 (5.1%) | 6318 (4.5%) | |
| Four or more visits | 3812 (1.3%) | 2180 (1.4%) | 1632 (1.1%) | |
| Charlson comorbidity index, n (%) | | | | 0.11606 |
| 0 | 240,435 (81.2%) | 128,128 (83.2%) | 112,307 (79.1%) | |
| 1 | 25,970 (8.8%) | 12,909 (8.4%) | 13,061 (9.2%) | |
| 2-3 | 8914 (3.0%) | 3939 (2.6%) | 4975 (3.5%) | |
| 4-5 | 912 (0.3%) | 360 (0.2%) | 552 (0.4%) | |
| >=6 | 19,741 (6.7%) | 8685 (5.6%) | 11,056 (7.8%) | |
| Autoimmune disease, n(%) | 24,873 (8.4%) | 12,683 (8.2%) | 12,190 (8.6%) | |
| Immunosuppressive treatments, n (%) | | | | |
| Selective immunosuppressants | 402 (0.1%) | 193 (0.1%) | 209 (0.1%) | −0.01271 |
| TNF-alfa inhibitors | 2521 (0.9%) | 1412 (0.9%) | 1109 (0.8%) | 0.01477 |
| Interleukin inhibitors | 599 (0.2%) | 302 (0.2%) | 297 (0.2%) | −0.00292 |
| Other immunosuppressants | 3314 (1.1%) | 1657 (1.1%) | 1657 (1.2%) | −0.00869 |
| Glucocorticoids | 26,975 (9.1%) | 13,241 (8.6%) | 13,734 (9.7%) | −0.03743 |
| Cytostatic | 721 (0.2%) | 316 (0.2%) | 405 (0.3%) | −0.01620 |

due to its effectiveness, narrow spectrum, and low cost[33], as recommended by the Swedish guidelines[3,6]. In a randomized controlled non-inferiority study, Skoog et al.[34] compared relapses and complication rates of two intervention groups treated with penicillin-V. The findings revealed no significant differences, suggesting the continued use of penicillin-V as a primary treatment for uncomplicated pharyngotonsillitis, with potential for shorter treatment durations. Furthermore, our study highlights that Sweden's restrictive approach to antibiotic use has demonstrated that limiting prescriptions does not increase complications, such as peritonsillar abscesses or rheumatic fever. This reinforces the safety and efficacy of penicillin-V as the preferred treatment. Additionally, the study found that complications were more frequent in patients receiving antibiotics compared those untreated, in line with previous studies[23,24]. Notably, however, patients treated with penicillin-V experienced fewer complications (1.6%) than those on other antibiotics (2.9%), echoing findings from a large retrospective study conducted in Israel[35]. A negative outcome control analysis

**Table 2 | Trends of antibiotic treatment (yes/no) and type of antibiotics (penicillin/other) among patients with pharyngotonsillitis who lived in Stockholm and Västra Götaland regions in Sweden and visited primary healthcare between 1 January 2018-31 December 2023**

| Period | Antibiotic treatment, n (%) | | Type of antibiotics[a], n (%) | |
|---|---|---|---|---|
| | Yes (n = 154,021) | No (n = 141,951) | Penicillin (n = 137,836) | Other antibiotic (n = 16,185) |
| Pre-pandemic | 81,504 (53.4) | 58,109 (41.6) | 72,739 (89.3) | 8765 (10.8) |
| Pandemic | 26,791 (39.9) | 40,369 (60.1) | 23,851 (89) | 2940 (11) |
| Post pandemic | 45,726 (51.3) | 43,473 (48.7) | 41,246 (90.2) | 4480 (9.8) |

Pre-pandemic: 1 Jan 2018-31 Jan 2020; Pandemic: 1 Feb 2020-28 Feb 2022; Post pandemic: 1 Mar 2022-31 Jan 2024.
[a]analysis only for those received antibiotic treatment.

**Table 3 | Proportion of complications observed within 30 days among patients with pharyngotonsillitis who lived in Stockholm and Västra Götaland regions in Sweden between 1 January 2018 and 31 Jan 2024, stratified by antibiotic treatment (yes/no) and by type of antibiotic treatment (penicillin/other antibiotics)**

| Complications | Antibiotics treatment n (%) | | Type of antibiotics[a] n (%) | |
|---|---|---|---|---|
| | Yes (n = 154,021) | No (n = 141,951) | Penicillin (n = 137,836) | Other antibiotic (n = 16,185) |
| Total | 2695 (1.75%) | 612 (0.43%) | 2230 (1.62%) | 465 (2.87%) |
| Type of complications | | | | |
| Peritonsillar abscess | 2647 (1.72%) | 577 (0.41%) | 2193 (1.59%) | 454 (2.81%) |
| Retropharyngeal and parapharyngeal abscess | 62 (0.04) | 38 (0.03) | 42 (0.03) | 20 (0.12%) |
| Rheumatic fever | 0 | 0 | 0 | 0 |
| Necrotizing fasciitis | <5 (0.0) | <5 (0.0) | <5 (0.0) | 0 (0.0) |
| Bacteremia | 17 (0.01) | 9 (0.006) | 15 (0.01) | <5 (0.0) |

[a]analysis only for those who received antibiotic treatment.

**Table 4 | Crude and adjusted odds ratio for the association between antibiotic treatment (No vs Yes) and type of antibiotic treatment (other antibiotics vs penicillin) with complications occurring within 30 days among pharyngotonsillitis patients in Stockholm and Västra Götaland regions in Sweden, overall and across different pandemic periods**

| Period | Antibiotic treatment (No vs Yes) | | Type of antibiotic[a] (Other vs Penicillin) | |
|---|---|---|---|---|
| | Crude OR (95%CI) | aOR[b] (95%CI) | Crude OR (95%CI) | aOR[b] (95%CI) |
| Overall | 0.24 (0.22−0.27) | 0.24 (0.22−0.26) | 1.80 (1.63−1.99) | 1.84 (1.66−2.04) |
| Pre-pandemic | 0.27 (0.24−0.32) | 0.27 (0.24−0.31) | 2.03 (1.75−2.36) | 2.05 (1.75−2.39) |
| Pandemic | 0.16 (0.13−0.19) | 0.16 (0.13−0.19) | 1.66 (1.35−2.03) | 1.71 (1.39−2.09) |
| Post pandemic | 0.26 (0.22−0.30) | 0.26 (0.22−0.30) | 1.62 (1.34−1.95) | 1.70 (1.40−2.06) |

aOR adjusted odds ratio, CI confidence interval.
[a]analysis only for those who received antibiotic treatment.
[b]adjusted for age, sex, sociodemographics, Charlson comorbidity index, vaccination status, primary care visits, and immunosuppressive treatments.

**Table 5 | Association between antibiotic treatment (No vs Yes) and type of antibiotic (other antibiotics vs penicillin) with specific complications within 30 days among pharyngotonsillitis patients in Stockholm and Västra Götaland regions in Sweden**

| Type of complications | Antibiotic treatment (No vs Yes) | | Type of antibiotic[a] (Other vs Penicillin) | |
|---|---|---|---|---|
| | Crude OR, 95% CI | aOR[b], 95% CI | Crude OR, 95% CI | aOR[b], 95% CI |
| Peritonsillar abscess | 0.23 (0.21−0.26) | 0.23 (0.21−0.26) | 1.79 (1.61−1.98) | 1.83 (1.65−2.03) |
| Retropharyngeal and parapharyngeal abscess | 0.66 (0.44−0.99) | 0.60 (0.40−0.90) | 4.06 (2.38−6.92) | 3.98 (2.32−6.82) |
| Rheumatic fever | — | — | — | — |
| Necrotizing fasciitis | 1.09 (0.22−5.38) | 0.85 (0.17−4.30) | — | — |
| Bacteremia | 0.57 (0.26−1.29) | 0.60 (0.26−1.35) | 1.13 (0.26−4.97) | 1.00 (0.22−4.52) |

aOR adjusted odds ratio, CI confidence interval.
[a]analysis only for those who received antibiotic treatment.
[b]adjusted for age, sex, sociodemographics, Charlson comorbidity index, vaccination status, primary care visits, and immunosuppressive treatments.

**Table 6 | Associations between antibiotic treatment (No vs Yes) and type of antibiotic (other antibiotics vs /penicillin) with complications occurring within 30 days among patients with pharyngotonsillitis in Stockholm and Västra Götaland regions in Sweden, stratified by sex and age**

| | | Antibiotic treatment (no vs yes) | | Type of antibiotic[a] (other vs penicillin) | |
|---|---|---|---|---|---|
| | | Crude OR (95% CI) | aOR[b] (95% CI) | Crude OR (95% CI) | aOR[b] (95% CI) |
| Age group (years) | 12–24 | 0.29 (0.25−0.33) | 0.29 (0.25−0.34) | 2.11 (1.75−2.55) | 2.10 (1.71−2.50) |
| | 25–39 | 0.25 (0.22−0.29) | 0.25 (0.21−0.28) | 1.81 (1.54−2.13) | 1.83 (1.56−2.16) |
| | 40–69 | 0.19 (0.16-0.23) | 0.19 (0.15-0.23) | 1.56 (1.29-1.88) | 1.65 (1.37-2.00) |
| | 70+ | 0.17 (0.10−0.28) | 0.17 (0.10−0.29) | 1.95 (1.13−3.38) | 2.00 (1.14−3.50) |
| Sex | Female | 0.25 (0.22−0.28) | 0.25 (0.22−0.29) | 1.76 (1.54−2.02) | 1.76 (1.54-2.2) |
| | Males | 0.23 (0.20−0.26) | 0.23 (0.20−0.26) | 1.97 (1.69−2.29) | 1.96 (1.68−2.29) |

*aOR* adjusted odd ratio, *CI* confidence interval.
[a]analysis only for those received antibiotic treatment.
[b]adjusted for age, sex, sociodemographics, Charlson comorbidity index, vaccination status, primary care visits, and immunosuppressive treatments.

was considered but could not be implemented, as no suitable outcome met the required criteria—being unaffected by antibiotic treatment, comparable in ascertainment to complications, and sufficiently frequent for precise estimation. Instead, we performed timing-oriented sensitivity analyses extending exposure windows to address potential temporal misclassification and support the robustness of our findings. The sensitivity analysis using a 5-day exposure window yielded similar results, indicating that the main associations were robust and that later prescriptions likely represented low-risk cases without added clinical benefit.

Although antibiotics overall were associated with increased risk of complications within 30 days in pharyngotonsillitis patients (likely due to selection of high-risk patients for treatment), this study found that other antibiotics (broad-spectrum antibiotics) were associated with higher odds of complications compared to penicillin-V, across all age groups and sexes. The findings suggest that penicillin-V is effective and a safe option to use to prevent complications compared to broader spectrum antibiotics. Broad-spectrum antibiotics, typically used when penicillin fails, may disrupt a broader range of bacteria, potentially increasing the risk of adverse effects and complications. These findings underscore the benefits of selecting narrow-spectrum antibiotics like penicillin-V when appropriate to reduce risk.

One of the primary strengths of this study is its large sample size, which includes data spanning five years from two major Swedish regions, illustrating different periods for evaluating trend in antibiotic prescription for pharyngotonsillitis patients and complications. This extensive dataset enhances the generalizability of the findings to broader populations. Additionally, the study accounts for various confounding factors such as sociodemographics, comorbidities (Charlson Comorbidity Index and immunosuppressive conditions), COVID-19 vaccination status, and healthcare utilization, enabling a more precise assessment of the association between antibiotic use and the occurrence of complication in pharyngotonsillitis patients. Another strength is its temporal analysis, covering pre-pandemic, pandemic and post-pandemic periods, which shed light on healthcare practices, particularly on antibiotic treatment in pharyngotonsillitis patients. The consistent use of penicillin-V as the primary treatment across all periods further underscores its effectiveness and low risk of complications in managing pharyngotonsillitis, even within a challenging healthcare environment.

Despite its strengths, the study has several limitations. First, as an observational study, there is a risk of confounding by indication. This occurs when the decision to prescribe antibiotics is influenced by factors such as the severity of illness or underlying health conditions, which are also related to the risk of complications. As a result, patients who receive antibiotics inherently are likely to have a higher baseline risk of developing complications compared to those who do not, irrespective of the treatment itself. Although we adjusted for sociodemographic characteristics, comorbidities, primary care visits, and vaccination status, residual confounding in this situation is likely, and should be considered when interpreting the causal relationship between antibiotic

treatment (vs. no treatment) and complication risk. Imbalances in age and comorbidity burden between treated and untreated patients are consistent with confounding by indication, as older and more comorbid patients were less often prescribed antibiotics but more likely to develop complications. Although we adjusted for these factors in multivariable models, residual confounding cannot be fully excluded and may partly explain the strong protective associations observed. Our sensitivity analyses, including alternative exposure definitions and timing restrictions, showed similar patterns, which supports the robustness of the findings but does not eliminate the possibility of bias. Future studies using designs that better emulate randomization (e.g., propensity score matching, instrumental variable approaches, or trial emulation frameworks) would be valuable to further address this issue.

By design, this study cannot establish causality; it only demonstrates associations observed in real-world practice. This limits the extent to which practical clinical or policy conclusions can be drawn. The available register data do not capture the reasons behind prescribing decisions or other clinical details, which means we cannot fully separate the effect of antibiotics from the underlying indications for their use. Nor could we apply additional approaches, such as negative control analyses, that might help to further assess this bias. Another limitation is the lack of detailed information on patient adherence to prescribed antibiotic regimens. Non-adherence, such as patients not completing their full course of antibiotics or improperly timing doses, could have influenced the study outcomes[36]. This factor is particularly relevant for those who received antibiotics yet still developed complications, as inadequate adherence may reduce the effectiveness of the medication, potentially leading to unresolved infections or the development of complications. Classifying antibiotic exposure within a defined period after the index diagnosis may introduce a degree of immortal time bias; however, given the short exposure window and study context, any such bias is expected to be minor. Lastly, reliance on healthcare records may result in missing or misclassified data. For instance, complications could be under-reported if patients sought care outside the documented healthcare system or if minor complications were not thoroughly recorded, potentially underestimating the true complication rates. We acknowledge that the number of primary care visits was measured after the index date and could be influenced by exposure or outcome, which may introduce bias.

Another limitation concerns the potential under-coding of antibiotic indications in healthcare registers. Evidence from English primary care shows that a substantial proportion of antibiotic prescriptions were not linked to an infection code, and this issue became more pronounced during the COVID-19 lockdowns[37]. Although Swedish registers are generally regarded as highly complete and reliable, similar gaps cannot be ruled out. Such under-coding could result in misclassification of pharyngotonsillitis episodes as untreated when antibiotics were in fact prescribed, which may have led to either underestimation or overestimation of complication risks. The findings should therefore be interpreted with this caveat in mind.

This study highlights penicillin-V's continued role as the preferred treatment option for pharyngotonsillitis, primarily due to its association with lower complication rates compared to other antibiotics. The results provide valuable evidence to guide clinicians, especially when faced with uncertainty between opting for penicillin-V or alternative treatments. Additionally, the notable decline in antibiotic prescriptions observed during the pandemic, followed by a rebound in the post-pandemic period, indicates a substantial shift in healthcare delivery and patient behavior. This pattern suggests that factors such as reduced healthcare access, heightened infection control measures, or a more cautious approach to prescribing during the pandemic likely influenced antibiotic use. The subsequent return to pre-pandemic prescribing levels highlights the importance of continued efforts in antibiotic management to prevent overuse and ensure optimal patient outcomes.

## Conclusion

This study demonstrates that the COVID-19 pandemic impacted antibiotic prescribing practices for pharyngotonsillitis in Swedish primary care, leading to a notable decline in overall antibiotic use during this period. Penicillin-V's continued use as a first-line therapy appears well-justified due to its association with a lower risk of complications. Additionally, the consistently lower complication rates observed among patients who did not receive antibiotics highlight the importance of careful antibiotic prescribing. These findings emphasize the necessity of appropriate antibiotic use and the continued monitoring of prescribing trends, particularly during the periods of healthcare disruption, such as the COVID-19 pandemic. Despite Sweden's low antibiotic use, the study found no evidence of underprescribing.

## Data availability

The study uses pseudonymised individual-level data from Swedish health registers, which cannot be shared openly under Swedish law. Access to the data may be granted by the responsible register authorities after approval from an ethics review board and completion of the required legal and data protection procedures. Information on how to apply for access to Swedish register data is available from Statistics Sweden (https://www.scb.se/en/statistical-database/) and National Board of Health and Welfare (https://www.socialstyrelsen.se/en/statistics-and-data/registers/).

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

## Acknowledgements

The SCIFI-PEARL project and this study has basic fundings from the Swedish government and the county councils, the ALF agreement (grants ALFGBG-938453, ALFGBG-971130, ALFGBG-978954 and ALFGBG1006729), the Swedish Research Council for Health, Working Life and Welfare (Forte), grant 2024-01711, and previously a joint grant from Forte and the Swedish Research Council for Environment, Agricultural Sciences and Spatial Planning (Formas), grant 2020-02828.

## Author contributions

A.S.: contributed to the literature research, conceptualisation, methodology, analyses, writing and critical revision of the manuscript. J.C., E.D., A.L., M.F., A.T., R.L., F.N.: contributed to conceptualisation, methodology, and critical revision of the manuscript. All co-authors reviewed and approved the final manuscript.

## Funding

## Competing interests

The authors report the following financial and non-financial interests that could be perceived as competing interests: F.N. owns shares in AstraZeneca. A.S. and F.N. participate in research projects funded by Bayer and AstraZeneca (regulator-mandated phase IV study; investigator-initiated study), with funding paid to their employer (no personal remuneration) and unrelated to the present work. A.L., M.F. and R.L. are employees of the Swedish Medical Products Agency, and J.C. and A.T. are employees of the Public Health Agency of Sweden. E.D. was employed at the Swedish Medical Products Agency at the time the study was conducted but has since changed employment. The views expressed in this article do not necessarily reflect those of their respective agencies.

## Additional information

¹School of Public Health and Community Medicine, Institute of Medicine, Sahlgrenska Academy, University of Gothenburg, Gothenburg, Sweden. ²Public Health Agency of Sweden, Solna, Sweden. ³Division of Use and Information, Swedish Medical Products Agency, Uppsala, Sweden. ⁴Karolinska Institutet, Department of Clinical Science and Education, Södersjukhuset, Stockholm, Sweden. ⁵Division of Licensing, Swedish Medical Products Agency, Uppsala, Sweden. ⁶Umeå University, Department of Clinical Microbiology, Umeå, Sweden. ⁷Karolinska Institutet, Department of Medicine Solna, Stockholm, Sweden. ⁸Karolinska Institutet, Institute of Environmental Medicine, Stockholm, Sweden. ✉e-mail: ailiana.santosa@gu.se

