## [Transparent Peer Review file · Communications Medicine]

Primary Care Pharyngotonsillitis Complications Following Absent or Deferred Antibiotic Treatment Across the COVID 19 Pandemic

Corresponding Author: Dr Ailiana Santosa

Version 0:

Reviewer comments:

Reviewer #1

(Remarks to the Author)

The authors examine antibiotic prescribing for pharyngotonsillitis in two large Swedish regions (2018–2023) and the 30-day risk of complications. Using register linkage, they compare treated versus untreated episodes and penicillin-V versus other antibiotics, finding that penicillin-V remains safe and that complication rates are lower than with broader-spectrum agents. The study is methodologically sound and clearly presented, offering clinically meaningful evidence that penicillin-V is a safe first-line therapy for pharyngotonsillitis. The limitations section appropriately addresses confounding by indication and other biases.

Specific comments:

Results

Table 2 summarises antibiotic prescribing pre-, during and post-pandemic, but a time-series plot of monthly prescribing rates would more clearly illustrate how utilisation evolved across the entire study period.

Discussion

Because your analysis relies on linking dispensed antibiotics to pharyngotonsillitis episodes, it would strengthen the Discussion to acknowledge potential under-coding of indication. For instance, <https://aricjournal.biomedcentral.com/articles/10.1186/s13756-023-01280-6> found that large antibiotic prescriptions in English primary care lacked an infection code—a proportion that worsened during COVID-19 lockdowns. Although Swedish registers are generally comprehensive, similar gaps could misclassify treated versus untreated episodes and slightly bias your complication-risk estimates. Please add a brief note of this limitation.

Reviewer #2

(Remarks to the Author)

Thank you for the opportunity to me for reviewing this manuscript investigating the relationship between antibiotics usage on pharyngotonsillitis and the following complications. Besides, the trend of antibiotics use for pharyngotonsillitis was also reported. All the above analyses were conducted before, during and after the COVID pandemic.

The author found there was a decrease in antibiotic use during the COVID-19 period, and then a rebound was observed. Besides, over the whole period, penicillin-V was kept on the most prescribed antibiotics. With logistic regression, they found strong protection against complications with non-users of antibiotics.

The topic is important and intriguing to me. However, several revisions or considerations are needed to strengthen the manuscript before publication. Below, I outline major and minor revisions intended to encourage the authors and help strengthen a future iteration of this work.

Major Comments

- Study Design & analysis: Please make it clear if this manuscript is a cohort or case-control. In Page 3, line 133, the author

indicates that it is a cohort study. However, the results were reported with odds ratio and adjusted odds ratio. From my understanding, the design appears cohort-based (following patients from exposure to outcome), not case-control. Therefore, presenting Risk Ratios (RR) is likely more appropriate than Odds Ratios (OR).

- I understand that no antibiotics or delayed antibiotics are sometimes recommended for self-limiting infectious diseases. However, in the clinics, there was still a higher proportion of people (more than 76.9% in hospital and 88.9% in A&E) with antibiotics for pharyngotonsillitis.[PMID: 40142520] In another Swedish study about pharyngotonsillitis, the proportion of receiving antibiotics is higher (10748 vs 3033) with a RADT obtained on the same day[PMID: 40142520]. In the current study, only half of the patients will choose antibiotics within the first 2 days after the index date.

1. Did the author do some matching steps? It would be great if the author could provide a flowchart of patient inclusion and exclusion criteria.

2. May we have the SMD between the two groups in the tables to illustrate whether there is a confounder by indication?

3. Could you please conduct a sensitivity analysis by expanding the exposure assessment window from 2 days to 5 days to check the robustness of the results?

- The estimation of 0.24 is way too protective from a statistical aspect. If we change the reference group, the estimation shows a 4 times risk of complications for the patients with antibiotics use. I am not sure if the results are robust enough, some sensitivity or negative outcome control analysis may be helpful.

Minor

- Clarify terminology: Define the term "deferred" antibiotics used in the manuscript.

- The exposure definition after the index may causing immortal time bias. But in this circumstance, I believe it is minor.

- While the immortal time bias risk might be minor, consider analyzing complication risk using time-to-event methods (a cumulative incidence plot will be better) instead of logistic regression. Logistic regression is also acceptable.

- "The number of primary care visits from the index date to outcome occurrence" looks like a covariate after the index date. Including such variables in the model may cause bias since the use of antibiotics or complications may impact such variable. If the author wants to include medical utilization as a proxy of health severity, a visit number within a specific time before index data may be more appropriate since it won't be affected by the exposure or outcome of interests.

- I am not sure about the main purpose of the second paragraph, which describes the overall decline trend of infections and antibiotics prescribed recorded in the literature. Does the other want to emphasize such a change of the prescription pattern may impact the prognosis of pharyngotonsillitis?

- The other minor issue is how to define the different COVID periods. The anchors is the occurrence of pharyngotonsillitis or the occurrence of any complications? It may not change the estimation, but it will be great to clarify.

- Page 4, line 252. No need "(aOR) "

Version 1:

Reviewer comments:

Reviewer #2

(Remarks to the Author)

All of my comments have been addressed by the author. Great work.

Response letter for

"The Effect of Absent or Deferred Antibiotic Treatment on Pharyngotonsillitis Complications in Primary Care Before, During, and After the COVID-19 pandemic"

Reviewers' comments:

Reviewer#1:

The authors examine antibiotic prescribing for pharyngotonsillitis in two large Swedish regions (2018–2023) and the 30-day risk of complications. Using register linkage, they compare treated versus untreated episodes and penicillin-V versus other antibiotics, finding that penicillin-V remains safe and that complication rates are lower than with broader-spectrum agents. The study is methodologically sound and clearly presented, offering clinically meaningful evidence that penicillin-V is a safe first-line therapy for pharyngotonsillitis. The limitations section appropriately addresses confounding by indication and other biases.

Response: We thank for the reviewer's supportive comments and are pleased that the study's design, clarity, and clinical relevance were well received. We particularly appreciate the recognition that the limitations section appropriately addresses confounding by indication, which reinforces the study's contribution to evidence on penicillin-V as a safe first-line therapy for pharyngotonsillitis.

Specific comments:

Results

Table 2 summarises antibiotic prescribing pre-, during and post-pandemic, but a time-series plot of monthly prescribing rates would more clearly illustrate how utilisation evolved across the entire study period.

Response: Thank you for this helpful suggestion. We have now included a time-series plot of monthly antibiotic prescribing rates in the supplementary material (*Suppl Figure 1*). This figure supports the results presented in the main text and provides a clearer illustration of utilisation trends across the full study period.

Results: Monthly prescribing trends (Suppl Figure 1) confirm this pattern, showing clear seasonal peaks pre-2020, a sharp decline at the onset of the COVID-19 pandemic, and only a partial rebound from 2022 onwards. (page 5, line 226-228)

Discussion

Because your analysis relies on linking dispensed antibiotics to pharyngotonsillitis episodes, it would strengthen the Discussion to acknowledge potential under-coding of indication. For instance, <https://aricjournal.biomedcentral.com/articles/10.1186/s13756-023-01280-6> found that large antibiotic prescriptions in English primary care lacked an infection code—a proportion that worsened during COVID-19 lockdowns. Although Swedish registers are generally comprehensive, similar gaps could misclassify treated versus untreated episodes and slightly bias your complication-risk estimates. Please add a brief note of this limitation.

Response: Thank you for your suggestion. Now we have added in the discussion under limitation (page 8, line 381-388).

“Another limitation concerns the potential under-coding of antibiotic indications in healthcare registers. Evidence from English primary care shows that a substantial proportion of antibiotic prescriptions were not linked to an infection code, and this issue became more pronounced during the COVID-19 lockdowns (36). Although Swedish registers are generally

regarded as highly complete and reliable, similar gaps cannot be ruled out. Such under-coding could result in misclassification of pharyngotonsillitis episodes as untreated when antibiotics were in fact prescribed, which may have led to either underestimation or overestimation of complication risks. The findings should therefore be interpreted with this caveat in mind."

Reference:

37. Yang, YT., Zhong, X., Fahmi, A. *et al.* The impact of the COVID-19 pandemic on the treatment of common infections in primary care and the change to antibiotic prescribing in England. *Antimicrob Resist Infect Control.* 12, 102 (2023). <https://doi.org/10.1186/s13756-023-01280-6>

Reviewer #2 (Remarks to the Author):

Thank you for the opportunity to me for reviewing this manuscript investigating the relationship between antibiotics usage on pharyngotonsillitis and the following complications. Besides, the trend of antibiotics use for pharyngotonsillitis was also reported. All the above analyses were conducted before, during and after the COVID pandemic. The author found there was a decrease in antibiotic use during the COVID-19 period, and then a rebound was observed. Besides, over the whole period, penicilin-V was kept on the most prescribed antibiotics. With logistic regression, they found strong protection against complications with non-users of antibiotics.

The topic is important and intriguing to me. However, several revisions or considerations are needed to strengthen the manuscript before publication. Below, I outline major and minor revisions intended to encourage the authors and help strengthen a future iteration of this work.

Response: We sincerely thank the reviewer for taking the time to evaluate our manuscript and for recognizing the importance of this topic. We appreciate the positive remarks on the relevance of studying antibiotic use for pharyngotonsillitis and related complications, as well as on the analysis of prescribing trends across the pre-, during, and post-COVID-19 periods. We also thank the reviewer for the constructive feedback and valuable suggestions. We have carefully considered all comments and revised the manuscript accordingly to strengthen its clarity, methodological transparency, and interpretation.

Major Comments

- **Study Design & analysis:** Please make it clear if this manuscript is a cohort or case-control. In Page 3, line 133, the author indicates that it is a cohort study. However, the results were reported with odds ratio and adjusted odds ratio. From my understanding, the design appears cohort-based (following patients from exposure to outcome), not case-control. Therefore, presenting Risk Ratios (RR) is likely more appropriate than Odds Ratios (OR).

Response: We thank the reviewer for this important point. Our study is a retrospective, register-based cohort design with 30-day follow-up after the index diagnosis. While risk ratios (RRs) are often considered more intuitive in cohort studies, the complication outcomes in our material were rare (<2%). In such circumstances, odds ratios (ORs) from logistic regression are numerically very close to RRs, and therefore provide equally valid effect estimates. Logistic regression was chosen because it is a robust and widely applied method for binary outcomes in register-based epidemiology, particularly when modeling rare complications. To ensure clarity, we have revised the Methods to explicitly describe the cohort design and the rationale for reporting ORs.

Methods: Given the low incidence of complications (<2%), ORs were considered appropriate as they closely approximate risk ratios in this setting (29). (page 5, line 195-196)

• I understand that no antibiotics or delayed antibiotics are sometimes recommended for self-limiting infectious diseases. However, in the clinics, there was still a higher proportion of people (more than 76.9% in hospital and 88.9% in A&E) with antibiotics for pharyngotonsillitis.[PMID: 40142520] In another Swedish study about pharyngotonsillitis, the proportion of receiving antibiotics is higher (10748 vs 3033) with a RADT obtained on the same day[PMID: 40142520]. In the current study, only half of the patients will choose antibiotics within the first 2 days after the index date.

1. Did the author do some matching steps? It would be great if the author could provide a flowchart of patient inclusion and exclusion criteria.

Response: We thank the reviewer for this insightful comment. We acknowledge that antibiotic prescribing rates for pharyngotonsillitis vary across healthcare settings and study designs. Unlike hospital- or A&E-based studies reporting higher prescribing rates, our analysis focused on primary care, where Swedish guidelines emphasize diagnostic testing and restrictive antibiotic use. No matching procedure was performed. Instead, we used multivariable regression to adjust for a wide range of potential confounders, including age, sex, sociodemographic factors, comorbidities, vaccination status, and healthcare utilization. Given the large sample size and comprehensive adjustment, we believe this approach is sufficient and preferable to matching, which would have reduced the study population without clear additional benefit. Patient selection has been described in detail in the Methods section, including the study population (individuals ≥ 12 years with a primary care visit), the application of a 180-day washout to identify unique index episodes, and the follow-up window for complications.

2. May we have the SMD between the two groups in the tables to illustrate whether there is a confounder by indication?

Response:

We thank the reviewer for this helpful suggestion. We have now added standardized mean differences (SMDs) for all baseline characteristics (Table 1). Most covariates showed negligible imbalance (SMD < 0.1). Moderate imbalance was observed for age (SMD = 0.13–0.17) and comorbidity burden (SMD = 0.11–0.12), indicating that untreated patients were slightly older and had a higher comorbidity burden, consistent with confounding by indication. Corresponding descriptions have been added in the Methods, Results, and Discussion sections to clarify both the rationale and interpretation of these findings.

Method: Standardized mean differences (SMDs) were calculated for each baseline characteristic to quantify differences between antibiotic-treated and untreated groups; values ≥ 0.1 were considered to indicate meaningful imbalance. (page 4, line 190-192)

Result: Most baseline characteristics were well balanced (SMD < 0.1), with moderate imbalances only for age (SMD = 0.13–0.17) and comorbidity burden (SMD = 0.11–0.12), indicating that untreated patients were somewhat older and had more comorbidities. (page 5, line 218-220)

Discussion: Imbalances in age and comorbidity burden between treated and untreated patients are consistent with confounding by indication, as older and more comorbid patients were less often prescribed antibiotics but more likely to develop complications. Although we

adjusted for these factors in multivariable models, residual confounding cannot be fully excluded, and may partly explain the strong protective associations observed. Our sensitivity analyses, including alternative exposure definitions and timing restrictions, showed similar patterns, which supports the robustness of the findings but does not eliminate the possibility of bias. Future studies using designs that better emulate randomization (e.g., propensity score matching, instrumental variable approaches, or trial emulation frameworks) would be valuable to further address this issue. (page 8, line 352-360)

3. Could you please conduct a sensitivity analysis by expanding the exposure assessment window from 2 days to 5 days to check the robustness of the results?

Response:

We thank the reviewer for this valuable suggestion. As requested, we conducted a sensitivity analysis extending the exposure window for antibiotic dispensing from 2 to 5 days after the index diagnosis of pharyngotonsillitis. The number of antibiotic prescriptions increased slightly (from 154,021 [52.0%] to 157,856 [53.3%]), whereas the proportion of complications remained largely unchanged. The association between antibiotic treatment and 30-day complications remained robust, with an even stronger protective effect (aOR = 0.11; 95% CI: 0.09–0.12). (Suppl Table 7). We observed a more pronounced protective association under the 5-day exposure definition. This likely reflects immortal-time bias, as patients classified as treated must remain free of complications until the antibiotic is dispensed, whereas early complications are attributed to the untreated group. These results are presented in the Supplementary Material (Supp Table S6 and S7) and are now referenced in the *Methods*, *Results*, and *Discussion* sections.

Method: We conducted a sensitivity analysis extending the exposure window for antibiotic prescriptions from 2 to 5 days after the index diagnosis to assess the robustness of the exposure definition. (page 5 line 201-203).

Results: Results from sensitivity analyses showed extending the exposure window to 5 days slightly increased the number of antibiotic prescriptions (from 52.0% to 53.3%) (Supp. Table 7), while complication rates remained unchanged, confirming the robustness of the main findings. (Supp. Table 8). (page 6, line 279-281)

Discussion: The sensitivity analysis using a 5-day exposure window yielded similar results, indicating that the main associations were robust and that later prescriptions likely represented low-risk cases without added clinical benefit. (page 7, line 321-323).

- The estimation of 0.24 is way too protective from a statistical aspect. If we change the reference group, the estimation shows a 4 times risk of complications for the patients with antibiotics use. I am not sure if the results are robust enough, some sensitivity or negative outcome control analysis may be helpful.

Response: We thank the reviewer for this important observation. The adjusted odds ratio of 0.24 reflects the association between antibiotic treatment (yes vs. no) and complications within 30 days after pharyngotonsillitis, and we agree that the magnitude appears strongly protective. To evaluate the robustness of this finding, we conducted a sensitivity analysis extending the exposure window from 2 days to 5 days, which increased the proportion of patients classified as treated. This yielded an adjusted odds ratio of 0.11 (95% CI: 0.09–0.12), a even more protective estimate that we attribute to immortal-time bias, as patients must remain event-free until treatment is initiated, whereas early complications accrue to the untreated group.

We also considered a negative outcome control, however, no outcome in our data met the necessary criteria- specifically, being unaffected by antibiotic treatment, recorded with comparable ascertainment to complications, and sufficiently frequent for precise estimation. Candidate outcomes were either too rare or prone to differential detection among treated patients. Instead, we performed timing-oriented sensitivity analyses (extending exposure windows) that address potential temporal misclassification. We believe this sensitivity analysis provides a robustness assessment of bias and supports the validity of our findings. We added this in the Discussion (page 7 line 316-323).

A negative outcome control analysis was considered but could not be implemented, as no suitable outcome met the required criteria—being unaffected by antibiotic treatment, comparable in ascertainment to complications, and sufficiently frequent for precise estimation. Instead, we performed timing-oriented sensitivity analyses extending exposure windows to address potential temporal misclassification and support the robustness of our findings. The sensitivity analysis using a 5-day exposure window yielded similar results, indicating that the main associations were robust and that later prescriptions likely represented low-risk cases without added clinical benefit.

Minor

- Clarify terminology: Define the term "deferred" antibiotics used in the manuscript.

Response: We appreciate the reviewer's valuable feedback.

We have now defined the term "*deferred antibiotics*" in the Methods section.

We defined deferred antibiotics if prescriptions dispensed between days 3 and 7 after the index diagnosis. (Page 4, line 174-175)

- The exposure definition after the index may causing immortal time bias. But in this circumstance, I believe it is minor.

Response: We agree that defining exposure after the index date may introduce a potential for immortal time bias. As the reviewer notes, in this context the impact is likely to be minor given the short time window and outcome under study. We have added a statement in the Discussion to acknowledge this potential limitation.

Classifying antibiotic exposure within a defined period after the index diagnosis may introduce a degree of immortal time bias; however, given the short exposure window and study context, any such bias is expected to be minor. Page 8, line 377-379)

- While the immortal time bias risk might be minor, consider analyzing complication risk using time-to-event methods (a cumulative incidence plot will be better) instead of logistic regression. Logistic regression is also acceptable.

Response: We thank the reviewer for this valuable suggestion. We agree that time-to-event analyses, such as survival models with cumulative incidence plots, can provide additional insights and help mitigate immortal time bias. However, the timing of complication outcomes in our dataset—recorded at the time of healthcare visits rather than symptom onset—does not allow for precise time-to-event estimation within the 30-day follow-up. We therefore used logistic regression to ensure consistency with previous work on pharyngotonsillitis complications and to provide interpretable effect estimates. Given the short follow-up period and low number of events, we consider any potential immortal time bias to be minimal.

- "The number of primary care visits from the index date to outcome occurrence" looks like a

covariate after the index date. Including such variables in the model may cause bias since the use of antibiotics or complications may impact such variable. If the author wants to include medical utilization as a proxy of health severity, a visit number within a specific time before index data may be more appropriate since it won't be affected by the exposure or outcome of interests.

Response: We thank the reviewer for this comment. We included the number of visits after the index date to reflect contemporaneous healthcare-seeking behavior, which we considered a relevant proxy for underlying health status. We acknowledge that this approach may introduce some bias, but given the short follow-up period and the primary role of this variable as an adjustment factor, we believe the impact on our estimates is minimal. We added in the discussion (page 8 line 383-484).

We acknowledge that the number of primary care visits was measured after the index date and could be influenced by exposure or outcome, which may introduce bias.

- I am not sure about the main purpose of the second paragraph, which describes the overall decline trend of infections and antibiotics prescribed recorded in the literature. Does the other want to emphasize such a change of the prescription pattern may impact the prognosis of pharyngotonsillitis?

Response: Thank you for the comment. The paragraph was intended to provide context for our study aim by summarizing broader pandemic-related changes in infection management and antibiotic prescribing. Our goal was not to suggest a direct effect on pharyngotonsillitis prognosis, but to highlight that these system-level shifts could influence trends in antibiotic use and complication risks observed in our analysis, underscoring the importance of interpreting these patterns within the broader healthcare context.

- The other minor issue is how to define the different COVID periods. The anchors is the occurrence of pharyngotonsillitis or the occurrence of any complications? It may not change the estimation, but it will be great to clarify.

Response: Thank you for pointing this out. The COVID-19 periods in our study were defined based on the occurrence of the pharyngotonsillitis episode (index date), not the occurrence of complications. We have clarified this definition in the Methods section of the revised manuscript.

The study period was divided into pre-pandemic (1 January 2018 - 31 January 2020, pandemic (1 February 2020 - 28 February 2022), and post-pandemic (1 March 2022 - 30 January 2024), defined according to the date of the index pharyngotonsillitis diagnosis. (page 4, line 161-163)

Page 4, line 252. No need "(aOR)"

Response:

Thank you for noticing this. We have removed "(aOR)" in the revised manuscript.